# Interaction between Amorphous Zirconia Nanoparticles and Graphite: Electrochemical Applications for Gallic Acid Sensing Using Carbon Paste Electrodes in Wine

**DOI:** 10.3390/nano10030537

**Published:** 2020-03-17

**Authors:** Chrys. O. Chikere, Nadimul Haque Faisal, Paul Kong-Thoo-Lin, Carlos Fernandez

**Affiliations:** 1School of Pharmacy and Life Sciences, Robert Gordon University, Garthdee Road, Aberdeen AB10 7GJ, UK; c.o.chikere@rgu.ac.uk (C.O.C.); p.v.s.kong-thoo-lin@rgu.ac.uk (P.K.-T.-L.); 2School of Engineering, the Robert Gordon University, Garthdee Road, Aberdeen AB10 7GJ, UK; N.H.Faisal@rgu.ac.uk

**Keywords:** amorphous materials, zirconium oxide nanoparticles, carbon paste electrode, phenolic acid, gallic acid, electrochemistry, cyclic voltammetry, differential pulse voltammetry

## Abstract

Amorphous zirconium oxide nanoparticles (ZrO_2_) have been used for the first time, to modify carbon paste electrode (CPE) and used as a sensor for the electrochemical determination of gallic acid (GA). The voltammetric results of the ZrO_2_ nanoparticles-modified CPE showed efficient electrochemical oxidation of gallic acid, with a significantly enhanced peak current from 261 µA ± 3 to about 451 µA ± 1. The modified surface of the electrode and the synthesised zirconia nanoparticles were characterised by scanning electrode microscopy (SEM), Energy-dispersive x-ray spectroscopy (EDXA), X-ray powdered diffraction (XRD) and Fourier-transform infrared spectroscopy (FTIR). Meanwhile, the electrochemical behaviour of GA on the surface of the modified electrode was studied using differential pulse voltammetry (DPV), showing a sensitivity of the electrode for GA determination, within a concentration range of 1 × 10^−6^ mol L^−1^ to 1 × 10^−3^ mol L^−1^ with a correlation coefficient of R^2^ of 0.9945 and a limit of detection of 1.24 × 10^−7^ mol L^−1^ (S/N = 3). The proposed ZrO_2_ nanoparticles modified CPE was successfully used for the determination of GA in red and white wine, with concentrations of 0.103 mmol L^−1^ and 0.049 mmol L^−1^ respectively.


**Highlights:**
For the first time, synthesised amorphous zirconium dioxide nanoparticles modified CPE has been used for the electrochemical determination of gallic acidInteraction of amorphous ZrO_2_ nanoparticles with graphite has been proposedAmorphous ZrO_2_ nanoparticles modified CPE has been used for the detection of gallic acid in wine samples.


## 1. Introduction

Nanoparticles and nanomaterials exhibit unique chemical, physical and electrochemical properties due to their very small sizes (between 1–500 nm), which are different from the bulk of the same compounds [1]. Based on these different properties, which include size, shape, crystal structure, surface area, surface energy, chemical composition, and catalytic properties, nanomaterials have been broadly applied in different analytical methods [2,3]. These include the production of new and improved electrochemical sensors and biosensors for specific electrochemical determination of analytes [4,5]. Chemically or physically modified electrodes using nanomaterials have been widely used for sensitive and selective analytical determination of important bioactive compounds such as GA [6,7,8]. The application of a carbon paste electrode modified with nanostructured materials for the determination of biological compounds has shown reasonable improvement on the electrochemical behaviors of compounds like ascorbic acid, dopamine, uric acid, and acetaminophen [9,10,11]. Carbon paste electrodes have been widely applied in electrochemical studies because of their easy preparation, low cost, and low background current, relative to solid graphite or noble metal electrodes. The ease at which they can be modified with different compounds, the simple surface renewal process, and the possibility for miniaturization make them good electrochemical sensors [12]. The main advantages of using nanoparticles-modified electrodes over macro-electrodes are their highly effective surface area, catalytic strength, control of the micro-environment, and mass transport [13,14].

Being a versatile compound, Zirconium oxide (ZrO_2_) has attracted considerable attention and has been used for several different practical applications. These include being used in fuel cell technology [15], its use as a catalyst or catalyst support [16], as a protective coating for optical mirrors and filters [17], and nano-electronic devices [18]. ZrO_2_ has been classified as a wide bandgap semiconductor that tends to be more conductive at increased temperatures [19]. Hence, the interest in its use as a modifier because of the high surface area, good biocompatibility, chemical inertness, good conductivity, and affinity for oxygen-containing groups [1,20]. Its physicochemical properties have led to its recent use as an electrochemical sensor in the determination of pesticides [21]; in a composite with carbon nanotube-modified gold electrodes for the detection of nitrites [22]. Furthermore, zirconium oxide nanoparticles supported on graphene oxide have been used for the electrochemical determination of dopamine and paracetamol in the presence of ascorbic acid [19]. Other interesting uses are zirconium dioxide-reduced graphene oxide composite for electrochemical sensing and biosensing of ascorbic acid, uric acid, and dopamine [23]; the creation of a novel class TiO_2_, and ZrO_2_ sol-gel modified carbon paste sensor for the detection of phenol by Hughes et al. [24]. All these uses of ZrO_2_ for electrochemical determinations have developed a wider interest in their use as nanoparticles for electrode modification. The effect of the possible interaction of the ZrO_2_ nanoparticles and the graphite powder used in this work is an advantage to their use as electrodes.

Gallic acid (2,3,4-trihydroxybenzoic acid) (see Figure 1) is a naturally occurring phenolic acid found in plants and an antioxidant with several biological activities. GA has shown to exhibit strong anticarcinogenic and antimutagenic activities [25,26,27]. As an antioxidant supplement in the human diet, it helps in reducing the risk of disease development by preventing or slowing down molecular oxidation in the human body [28], also known as oxidative stress. Furthermore, oxidative stress has been linked to human diseases like, diabetes, cancer, Parkinson’s Disease, cardiovascular diseases, and Alzheimer’s Disease [29,30]. With an increase in the importance of antioxidants in our diets, several analytical methods have been used for the determination of GA concentration or antioxidant capacity in plants, wine, or other matrices. These studies have been mainly done using spectrophotometric methods [31,32], chromatographic [33,34], and flow injection-chemiluminescence [35]. However, the main drawbacks of some of these methods are the cost of equipment, lengthy sample preparation procedures, and the use of toxic and expensive reagents, which could be detrimental to the environment. Hence, there is a need for alternative analytical methods to determine individual phenolic compounds like gallic acid.

Electrochemical based techniques [3,6,7,8,36,37], have recently been generating a lot of interest in gallic acid determination as they are fast, sensitive, inexpensive, portable, and require little or no sample pre-treatment. In addition, the electrochemical methods are capable of miniaturization and have no problem with colour interferences from samples, as is the case sometimes in spectrophotometric methods.

In this work and for the first time, amorphous zirconium oxide nanoparticles were used to modify carbon paste electrode for the sensitive electrochemical determination of gallic acid. The proposed method was found to be simple, rapid, and inexpensive. The characteristics of the modified electrode were also studied, and the electrode was used for the determination of gallic acid concentration in red and white wine.

## 2. Materials and Methods

### 2.1. Reagents

Zirconium hydroxide (Zr(OH)_4_) powder (M. wt. 159.25 g mol^−1^), zirconium (IV) oxide (M. Wt. 123.22 g mol^−1^; ≥99%), graphite powder, paraffin oil, analytical grade salts of potassium chloride, sodium dihydrogen phosphate dihydrate, disodium hydrogen phosphate heptahydrate, boric acid, glacial acetic acid, and gallic acid (anhydrous, M. Wt. 170.12 g mol^−1^) were purchased from Sigma Aldrich (London, UK). Analytical grade methanol, phosphoric acid (H_3_PO_4_), and sodium hydroxide were from Merck (Darmstadt, Germany). All the reagents were of the highest purity and no further purification was required. All the aqueous solutions used were freshly made with 18.2 MΩ.cm resistance value doubly distilled water. All stock solutions prepared were refrigerated at 4 °C and protected from light. The wine samples for the analytical application were commercially available brands of wine (Casillero Del Diablo Cabernet Sauvignon and Sauvignon Blanc).

### 2.2. Instrumentation and Measurement

Ivium vertex one potentiostat-galvanostat with Iviumsoft software (Eindhoven, Netherlands) was used for all the electrochemical measurements, which includes cyclic voltammetry (CV), differential pulse voltammetry (DPV), and electrochemical impedance spectroscopy (EIS). The potentiostat-galvanostat was connected to a standard three-electrode cell system. The amorphous zirconia nanoparticles modified carbon paste electrode in the solvent-resistant plastic casing was used as the working electrode. In the electrochemical cell, Ag/AgCl (saturated KCl) and the platinum electrodes worked as the reference and counter electrodes, respectively.

The morphology of the amorphous zirconia nanoparticles was investigated by scanning electrode microscopy (SEM) from Carl Zeiss variable pressure scanning electron microscope (Oberkochen, Germany). The microscope was fitted with Oxford instruments and an energy dispersive x-ray analysis (EDXA) system, which was used to perform elemental chemical analysis.

The size characterisation of the synthesised ZrO_2_ was done by studying the particle size of the nanoparticles using the Malvern Zetasizer Nano ZS (Malvern, UK).

X-ray diffraction (XRD) patterns were collected on a PANalytical Empyrean powder diffractometer (Loughborough, UK) equipped with a Cu Kα tube and a Johansson monochromator.

The Fourier transform infrared spectrometer used for the analysis of the chemical bonding found in the zirconia nanoparticles was the Thermo Scientific Nicolet iS50 FTIR Spectrometer (Waltham, MA, USA).

The measurements of pH conditions were carried out with a Fisher Scientific (Loughborough, UK) Mettler Toledo Benchtop pH meter. The buffer solutions pHs were moved from acidic to basic and vis versa using 0.01 mol L^−1^ phosphoric acid to reduce pH and 0.1 mol L^−1^ sodium hydroxide to increase the pH. All potentials quoted in the experiments carried out were relative to the Ag/AgCl reference electrode.

### 2.3. Synthesis of Zirconium Dioxide Nanoparticles

The synthesis of the ZrO_2_ nanoparticles was done using an adaptation of the hydrothermal synthesis method described by Pei et al. [38]. In brief, Zirconium hydroxide (Zr (OH)_4_, (0.25 g) was dissolved in deionised water (60 mL). The solution was then placed in a 100-mL synthesis autoclave reactor and the system maintained at 80 °C–180 °C for different times. The temperature of the reactor was increased from 80 °C to 180 °C for 2 h for every cycle. This was continued for 12 h, which was equivalent to 6 cycles. A white solid was generated, which was then filtered, washed with deionised water several times, and dried at 60 °C to yield a white powder. The morphology, elemental composition, and particle size of the ZrO_2_ nanoparticles were analysed by scanning electron microscopy (SEM), EDXA, and XRD, respectively.

### 2.4. Characterisation of the Synthesised ZrO_2_ Nanoparticles and Modified Electrode

SEM and EDXA were used to characterise the morphology and elemental composition of the synthesised ZrO_2_ nanoparticles. The ZrO_2_ nanoparticles samples (0.1 mg) were placed on a Zeiss screw mount covered with double-sided carbon tape, then put into the microscope for SEM and EDX analysis.

The room-temperature XRD measurements were carried out by recording data on a low-background silicon sample holder in the range 5° < 2 θ < 80°, with a step size of 0.013°. Data acquisition time was 50 min.

The size characterisation of the nanoparticles was carried out by suspending the ZrO_2_ nanoparticles in ethylene glycol at a concentration of 1 × 10^−3^ mol L^−1^. The suspension was then placed in a sonicator for 30 min and then measured using a Zetasizer.

FTIR measurement was done by placing the nanoparticles powder (10 mg) on the surface of the FTIR/ATR (attenuated total reflection) crystal. The samples were then analysed in the mid-infrared (MIR) range of 400–4000 cm^−1^ with a resolution of 4 cm^−1^ and 32 scans, set in transmittance mode.

### 2.5. Preparation of Buffers

The 0.1 mol L^−1^ phosphate buffer solution (PBS) was prepared by putting together the appropriate volumes of 0.2 mol L^−1^ Na_2_HPO_4_ and 0.2 mol L^−^^1^ NaH_2_PO_4_ to 50 mL, then diluted to 100 mL with deionised water.

The Britton-Robinson (BR) aqueous buffer solution was prepared by mixing the appropriate volumes of acids and basic buffer components, which includes phosphoric acid (0.04 mol L^−1^), boric acid (0.04 mol L^−1^), and acetic acid (0.04 mol L^−1^); which were then mixed with sodium hydroxide solution (0.2 mol L^−1^), when the basicity needed to be increased.

The acetate buffer was prepared by creating a 0.2 mol L^−1^ acetic acid, made by adding 1.5 mL of glacial acetic acid with 100 mL distilled water. It was then mixed with appropriate volumes of sodium acetate solution—made with 0.64 g of sodium acetate salt dissolved in 100 mL of distilled water (36.2 mL of sodium acetate solution plus 13.8 mL of acetic acid solution).

### 2.6. Preparation of the Electrode

The carbon paste electrode was prepared by mixing graphite powder (0.65 g) and paraffin oil (0.35g, ~0.2 mL) in a 65:35 ratio. The modified electrode was fabricated by combining graphite powder (0.6 g), amorphous zirconium oxide nanoparticles (0.1 g), and paraffin oil (0.3 g) in a ratio of 60:10:30 (*w*/*w*/*w*). The mixture was transferred to an agar mortar and thoroughly blended with a pestle. A control electrode was prepared with the same amount of graphite and paraffin oil, but with no ZrO_2_ nanoparticles. All the resulting pastes were then packed into solvent-resistant plastic cylindrical electrode casings (internal diameter of 2.87 mm). The other end of each electrode was then connected to a copper wire. Fresh electrode surfaces were made by smearing more paste, followed by polishing the face of the electrode on a filter paper.

### 2.7. Electrochemical Measurements

Cyclic voltammograms and differential pulse voltammograms were obtained from the electrochemical oxidations of GA on the surface of the carbon paste electrodes and the amorphous ZrO_2_-modified CPE. The analysis of GA on the electrode surfaces was carried out in phosphate buffer solution (0.1 mol L^−^^1^, pH 2.0) at room temperature. The CV measurements were scanned in a range of 0.0 V to +1.8 V at a scan rate of 100 mVs^−^^1^. Meanwhile, differential pulse voltammetry (DPV) were recorded at a potential range from 0.0 to +1.2 V with a pulse amplitude of 0.08 V and a pulse period of 0.2 s.

Electrochemical impedance spectroscopy provided information on the impedance changes of the different electrodes (bare CPE and the ZrO_2_ nanoparticles modified CPE surfaces), with their results shown in Nyquist plots. This EIS measurement was carried out in a frequency range of 100 KHz–0.01 Hz (50 points within the frequency range) and a potential of 0.4 V in 0.1 mol L^−1^ containing 5 mmol L^−1^ [Fe(CN)_6_]^3^^−^^/4−^ redox solution.

The influence of the accumulation time of GA (1 × 10^−3^ mol L^−1^) on the surface of the modified electrode was studied. Accumulation time is an important parameter that affects the amount of GA adsorbed on the surface of the electrode. The voltammograms produced by the GA determination, whilst increasing the accumulation times every minute up to eight minutes (0–480 sec), were recorded.

The reproducibility and repeatability of the method were studied using eight different modified electrodes to measure six GA replicates (10 mL, 1 × 10^−3^ mol L^−1^) in phosphate buffer (0.1 mol L^−1^, pH 2.0), using DPV.

The interference of foreign ions in the determination of GA was investigated by the addition of GA solution (5 mL of 1 × 10^−3^ mol L^−1^) in a solution that contained Ca^2+^, Fe^3+^, K^+^, and Na^+^ ions (5 mL of 0.1 mol L^−1^). In addition, caffeine and ascorbic acid (5mL of 0.1 mol L^−1^) were added to the GA solution (1 × 10^−3^ mol L^−1^), vigorously stirred for 2 min, and DPV recorded.

### 2.8. Preparation and Determination of Real Samples

The Gallic acid content in spiked (Red and White wine) samples were analysed using DPV. This was by recording voltammograms produced by the standard addition of known volumes and concentrations of GA. Real wine samples (1 mL) were added into the voltammetric cell and made up to 10 mL with phosphate buffer solution (0.1 mol L^−1^, pH 2.0). The samples were brought to a pH of 2.0 using 0.1 mol L^−1^ phosphoric acid. All samples were made in triplicates.

Aliquots (1 mL) of GA standard (1 × 10^−2^ mol L^−1^) were then added to the wine samples and vigorously stirred with a magnetic stirrer for two mins. After every addition, differential pulse voltammogram measurements were recorded. This was done to allow the recovery of the analyte.

## 3. Results and Discussion

### 3.1. Characterisation of the ZrO_2_ Nanoparticles

The ZrO_2_ nanoparticles modified CPE was characterised by first studying the sizes of the ZrO_2_ nanoparticles using a Zetasizer. The sizes were recorded and ranged between 20 to 470 nm, which is in line with other works in the literature, where their synthesis produced sizes ranging from 10 to 400 nm [39,40].

SEM provides information on the external morphology, chemical composition, and crystalline structure of the ZrO_2_ nanoparticles. Here, it showed the tetragonal-like structures of the ZrO_2_ nanoparticles with non-homogenous size distribution (Figure 2a,b). The particles did not appear consistently as discrete particles, but as small aggregations with average sizes ranging from 10 µm to 100 µm. The increased particle sizes observed in the SEM images could be attributed to any of these aggregation steps: (1) adsorption and aggregation of primary building units, small oligomer; (2) formation of thicker oligomers prior to interfacial attachment; (3) lateral growth of nanosheets via the addition of building units to edges; and (4) formation of vertical nanosheet stacks accompanied by lateral addition of building units [41]. However, when they are presented as discrete particles, on analysis by the Zetasizer, the sizes range averagely from 20 nm to about 470 nm. [42]. These micro-particle-like sizes of some of the ZrO_2_ are consistent with the XRD results, which shows that the compounds are in amorphous form. This agrees with the work of Opalinska et al. [39], where they found that the ZrO_2_ nanoparticles form aggregates at temperatures lower than annealing temperatures between 500 °C to 1100 °C. However, on increasing annealing temperatures, the nanoparticles become independent particles in the monoclinic phase.

The XRD pattern of the prepared ZrO_2_ nanoparticles as seen in Figure 3 reveals a high structural disorder at long-range, which is compatible with XRD patterns of amorphous ZrO_2_ [43]. The structures showed a broad peak corresponding to a reflection of tetragonal ZrO_2_ and the close broader peak corresponding to monoclinic ZrO_2_. The formation of the tetragonal shaped zirconia being the main phase of the nanoparticles could be because of the presence of hydroxide complex at the generation stage of the synthesis of [Zr (OH)_4_^+2^.H_2_O]_4_^+2^. (OH)^−8^ hydroxo-complex. This role played by the hydroxide complex in the formation of zirconia nanoparticles has previously been noted in the literature [44,45]. Here, it was suggested that the predominant tetragonal phase is attributed to the presence of NaOH in the generation stage of the hydroxo-complex.

Fourier Transform Infrared spectroscopy (FTIR) was also used for the characterisation of ZrO_2_ nanoparticles and recorded in a range of 400 cm^−1^ to 4000 cm^−1^ to confirm the different elements and bonding in the compound. The FTIR spectra, as seen in Figure 4, shows a dominant absorption band between 500–600 cm^−1^, which is attributed to the deformation of the Zr-O-Zr bond. Meanwhile, the peak centered around 1000 cm^−1^ and 1380 cm^−1^ are attributed to the stretching vibration of Zr-O. On the other hand, the bands around 1590 cm^−1^ and 1650 cm^−1^ are attributed to the bending vibrations of chemisorbed water molecules, as described by Prakashbhabu et al. [46]. Furthermore, the broad and strong peak at 3445 cm^−1^ is due to the adsorbed moisture on the surface of the compound, hence showing the O-H stretching of water.

### 3.2. Electrochemical Behaviour of GA At the Amorphous ZrO_2_-CPE

Cyclic voltammetry (CV) and differential pulse voltammetry (DPV) was used to study the voltammetric behaviour of GA. The cyclic voltammograms (Figure 5) show the determination of 1 × 10^−2^ mol L^−1^ gallic acid using a bare CPE, bare GCE, and ZrO_2_ nanoparticles-modified CPE in phosphate buffer of pH 2.0, at a scan rate of 100 mVs^−1^ in a potential range of 0.0 to 1.8 V at room temperature. Figure 5 shows the cyclic voltammograms of GA with bare CPE, ZrO_2_ nanoparticles modified CPE and bare glassy carbon electrode, in a 0.1 mol L^−1^ phosphate buffer of pH 2.0 at a scan rate of 100 mVs^−1^. The carbon paste electrodes were used on GA (1 × 10^−2^ mol L^−1^), where they produced peak currents of 451 ± 1.1 µA at a peak potential of 0.69 V for the ZrO_2_ nanoparticles modified CPE, 260 ± 3.2 µA at 0.64 V peak potential for bare CPE, and 309 ± 0.65 µA peak current at a peak potential of 0.67 µA when it was just a bare glassy carbon electrode. Looking at the positions of the anodic peaks, a strong positive shift is visible, as compared to other gallic acid determinations with peak currents at peak potentials of 0.5 V, 0.6 V, and 0.65 [8,47,48]. These could be a result of the pH of the measurements, scan rate of measurements, or the inability of the electrode to catalyse the reaction. With the peak potential having a positive shift, it can be noted that peak current enhancement was not based on electro-catalysis, but on increased electroactive surface area brought about by the nanoparticles. The amorphous ZrO_2_ nanoparticles used for the modification of CPE increased the electroactive surface area of the electrode about eightfold; from 0.0345 cm^2^ to 0.276 cm^2^. The positive shift of the peak potential produced by the zirconia nanoparticles modified electrode shows an increased overpotential, possibly as a result of the modification. If the reaction was electro catalysed or the modified electrode acted as an electrocatalyst, there would have been a reduction of peak potential. The reduction of peak potential normally suggests a faster reaction with less overpotential. Meanwhile, the lower the overpotential, the better is the catalyst.

From the voltammograms shown in Figure 5, it was deduced that although the bare CPE and bare glassy carbon electrode (GCE) oxidised the GA, the modified CPE produced comparatively better oxidation peak currents.

The gallic acid produced two oxidation peaks on the surface of the modified electrode (Figure 5a) in the course of the anodic sweep from 0.0 to 1.8 V. This is consistent with other cyclic voltammetric measurements of GA in the literature [7,49,50]. The voltammograms (Figure 5a) showed two anodic peaks on the forward sweep and currents of very low intensity with no visible peak at the return sweep. This is attributed to two consecutive oxidation reactions of GA during the anodic sweep and no detectable reduction reaction during the reverse sweep (Figure 5a and Figure 6).

Furthermore, both electrodes were used to measure the blank buffer (0.1 mol L^−1^ phosphate), to confirm that the buffer showed no current response at the potential range of 0.0–1.8 V (Figure 5a). All the other electrodes did not produce any oxidation as well. However, when GA was added to the phosphate buffer solution, all the electrodes produced anodic peaks, with the ZrO_2_ nanoparticles showing the highest peak current.

It is worth mentioning that the two peaks produced by the GA had previously been characterised in the literature as the formation of a semiquinone radical for the first peak, which then oxidised to a quinone in the second peak [7], while using SiO_2_ nanoparticles modified CPE. The oxidation at the first peak was shown to be that of the galloyl group, as can be seen in Figure 6. This then leads to the second peak, developed from the third hydroxyl (−OH) group found in the galloyl moiety of the gallic acid [7,49]. This is reinforced by the fact that the carboxylic group (-COOH) gets electro-oxidised at 2.0 V and has a CO_2_ bi-product [51], while the oxidation peaks of the GA, in this case, occur at peak potentials of 0.69 V and 0.85 V when using CV.

The EIS was used to investigate the difference in the behaviour of the bare CPE and the ZrO_2_ nanoparticles modified CPE. Both electrodes were used to measure [Fe (CN)_6_]^3−/4^^−^ (5 mmol L^−1^) redox solution, as can be seen by the Nyquist plots in Figure 5b. The Nyquist plots demonstrate the semicircle domain of the bare CPE and the modified CPE. They reflect the charge transfer resistance, attributed to the redox process of the [Fe (CN)_6_]^3−/4^ redox solution. The large semicircle portion exhibited by the bare CPE, corresponds to the high charge transfer resistance (R_ct_), associated with the highest capacitance [8]. Meanwhile, the ZrO_2_ nanoparticles modified CPE, reduces the barrier to interfacial electron transfer. The response shown by the electrodes using EIS corresponds to the Randles equivalent circuit seen in Figure 4b (inset).

Furthermore, the difference in the electrochemical response of the ZrO_2_ nanoparticles modified CPE and the bare CPE were further investigated, by using the electrodes to measure the redox couple [Fe(CN)_6_]^3−/4−^ (1.0 × 10^−3^ mol L^−1^), using CV at a scan rate of 100 mVs^−1^. From Figure 5c, one can see an increase in peak current with the modified CPE, when compared to the bare CPE.

On the other hand, the surface area of the amorphous ZrO_2_ nanoparticles-modified CPE was studied using [Fe(CN)_6_]^3^^−/4+^ redox solution as the analyte; and the results were recorded as can be seen in Figure 7. The redox solution [Fe(CN)_6_] (1.0 × 10^−3^ mol L^−1^) was measured using CV and the results recorded at different scan rates from 25 mVs^−1^ to 250 mVs^−1^. From the voltammograms in Figure 5c and 7, the reaction was found to be reversible. The reversible system was then analysed by using the Randles-Sevcik equation, to determine the surface area of the electrode as shown below:*Ip* = 2.69 × 10^5^*n*^3/2^*A D*^1/2^*v*^1/2^*C*_0_(1)
where *Ip* is the anodic peak current for [Fe(CN)_6_]^3−/4+^, *A* is the surface area of the electrode, *n* is the number electrons in the reaction (*n* = 1), *D* is the diffusion coefficient which is 7.6 × 10^−6^ cm^2^s^−1^, *ν* is the scan rate, and *C* is the concentration of [Fe(CN)_6_]^3−/4+^.

From the plot of the anodic peak current *Ip* against the square root of the scan rate (*ν*^1/2^), as can be seen in Figure 7b, a linear relationship between the two parameters was observed. Thus, the linear regression equation produced by the anodic peak currents was *Ipa* = 22.22 *ν*^1/2^−20.207 with *R*^2^ = 0.9980, while that of the cathodic peak currents was *Ipc* = 20.543 *ν*^1/2^−23.075 with *R*^2^ = 0.9980. Hence, the surface area calculated for the ZrO2 nanoparticles modified CPE was 0.276 cm^2^, higher than the surface areas of 0.0345 cm^2^ and 0.0648 cm^2^ for the bare CPE and bare GCE, respectively.

### 3.3. Effect of pH on GA Oxidation

The effect of buffer pH on the current response of the ZrO_2_ nanoparticles modified CPE was studied (Figure 8). This effect was studied using CV at ZrO_2_ nanoparticles modified CPE to measure GA (1 × 10^−2^ mol L^−1^, 0.1 mol L^−1^ phosphate buffer) at a pH range of 2.0 to 8.0 and scan rate of 100 mVs^−1^. The pH of the buffer in which GA was dissolved affects the electro-oxidation activity of GA on the surface of the modified CPE. Within the pH range of 2.0–8.0, the anodic peak current of GA showed a decrease with increasing pH using CV and an increasing negative shift of the peak potential from 0.64 to 0.38 V, as shown in Figure 8. The voltammograms showed a well-defined and shaped oxidation peak at a pH of 2.0, with the highest peak current of 374.49 µA, showing a clear influence of the pH on the electrochemical oxidation of GA at the ZrO_2_ nanoparticles modified CPE surface. Thus, a phosphate buffer solution of pH 2.0 was used for subsequent experiments in this work.

Further increase in the pH value of the solution to 10 showed no GA oxidation. The gradual negative shift of the peak potential as the pH increases showed a linear relationship between the anodic peak current and peak potentials of the different pH values. The relationship between the pH and the peak potential *E_p_* was studied and the graph produced in Figure 9. This relationship produced a linear regression equation of *E_pa_* = −46.314 pH + 471.24 with *R*^2^ = 0.9941, between pH 2.0 to pH 6.0, which produced a slope *E_pa_*/pH of the regression line of 46 mV/pH that can be compared to the Nernstian value 59 mV/pH at 25 °C. This suggests that the number of electrons and protons involved in the oxidation of GA was equal [51]. However, at pH 8.0, the peak current increased, which is in agreement with other literature where similar redox reaction with gallic acid at pH 8.0 was studied [52].

### 3.4. Effect of Modifier Concentrations on Gallic Acid Oxidation

The amount of ZrO_2_ nanoparticles blended in the graphite paste to obtain an optimised CPE for the gallic acid determination was studied (See SI). ZrO_2_ nanoparticles powder were weighed into the graphite powder at quantities of 0.5 g, 0.1 g, 0.15 g and 0.2 g, while adjusting the graphite powder appropriately to meet the required ratio. In this case, the ratios of graphite/ZrO_2_/paraffin (*w*/*w*/*w*), were 65:5:30, 60:10:30, 55:15:30, and 50:20:30, respectively. The modified electrodes were then used for the determination of 1 × 10^−3^ mol L^−1^ GA in a 0.1 mol L^−1^ phosphate buffer at a pH of 2.0, using DPV at a scan rate of 100 mVs^−1^. The peak currents produced by the 0.1 g ZrO_2_ nanoparticles mixtures were found to be the best. This might be ascribed to the disruption of higher concentrations of ZrO_2_ nanoparticles on the graphite-ZrO_2_ interaction and electron flow. Thus, for all the subsequently modified electrodes, 0.1 g ZrO_2_ nanoparticles were used.

### 3.5. Effect of Scan Rate on Gallic Acid Oxidation

The effect of scan rate on the electrochemical behaviour of GA (1 × 10^−3^ mol L^−1^) at the ZrO_2_ nanoparticles CPE at different scan rates ranging from 25 mVs^−1^–1500 mVs^−1^ was investigated using cyclic voltammetry. With the increase in scan rate and electro-oxidation of GA, there was a proportional increase in peak current and a positive shift in peak potential, as shown in Figure 10.

The increase in scan rate produces a positive shift in the peak potential and a linear increase in peak currents. This phenomenon suggests a kinetic limitation in the reaction of GA at the surface of the modified electrode [53]. Plotting the peak current (*I*_p_) against the scan rate (*v*), in the range of 25–1000 mVs^−1^, produced a straight line. This suggests that GA interacts with the ZrO_2_ nanoparticle CPE in an adsorption-controlled process. This created a linear regression equation; thus, *I_p_* (µA) = 0.0013 *v* (mVs^−1^) + 0.2211, *R*^2^ = 0.9956.

Plotting the relationship between the logarithm of peak current (log *I*_p_) and the logarithm of the scan rate (log *v*) in 25–1000 mVs^-1^ scan range also generated a straight line with a linear regression equation: −Log *I*_p_ (µA) = 1.7884 Log *v* (mV/s) + 2.7607 and R^2^ = 0.9902. These results indicate a diffusion and an adsorption-controlled reaction of GA at the ZrO_2_ nanoparticles modified CPE, as seen in Figure 10b.

### 3.6. Effect of Adsorption Time on the Modified CPE

The effect of adsorption time or accumulation time of GA on the modified CPE was investigated by the detection of GA (1 × 10^−3^ mol L^−1^, 0.1 mol L^−1^ phosphate buffer at pH 2.0) at room temperature with a scan rate of 100 mVs^−1^ using DPV. As seen in Figure 11, the voltammograms show an increase in peak current as time elapses. There is an increase in peak current after the first scan and subsequent scans after every minute to a maximum at the fifth minute. Further scans after the fifth minute show a decrease in peak current. The phenomenon can be attributed to the adsorption-controlled reaction confirmed by the scan rate, where the GA reaches a saturation point on the surface of the electrode. Thus, any further scan does not produce an increase in peak current, but a decrease. This is a result of electrode fouling, which is common with the production of phenoxy radicals and their subsequent reaction with phenol creating a polymeric adherent film that is deposited on the electrode surface [54]. This agrees with the work done by Tashkhourian [7] and Chikere et al. [8].

### 3.7. Amorphous Zirconia and Graphite Interaction

Amorphous zirconia in the monoclinic or tetragonal phase (Figure 3) is an oxide that is typically laced with hydroxyl species like other metal oxides [55], the latter commonly used as a catalyst and catalyst-supporting material. The tetragonal phase of zirconia does exist at low temperature, when they are synthesised from initially formed amorphous zirconia [56], as seen in the XRD pattern. The structure of the compound provides a versatile surface because of its lack of symmetrical lattice [57], with the possibilities of hydroxyls (Zr-OH), oxygen vacancies, coordinatively unsaturated Zr-O pairs, and Lewis acid sites (Zr^3+^, Zr^4+^) [58]. Furthermore, Terki et al. [59] proposed two different Zr-O bond distances and charge densities for the tetragonal Zirconia structure. They noticed that the Zr-O bond was covalent with some degree of ionic character; hence, the capability of electron transfer was noticed in the electrochemical determination of GA. It should also be noted that dissociative adsorption of water on monoclinic zirconia is exothermic, thus occurring at room temperature with hydroxylation at ambient temperatures [60]. On the other hand, graphite being held together with van der Waal dispersion forces show delocalized electrons moving around the sheet. The delocalization results in loosely bound π-electrons with very high mobility; hence, the role they play in the electrochemical properties of the graphite [61]. The mixing of zirconia and graphite (Figure 12) produces an interaction that enhances the peak current of the oxidised gallic acid. The physical mixing of the two electroactive compounds could have produced weak ionic bonds between the oxygen anions and the cations of the graphite sheet. With the adsorption of water on to the monoclinic zirconia and its inherently weak ionic character with delocalized electrons on the graphite sheets, weak ionic bonds are formed, as shown in Figure 12. Hence, the ZrO_2_-carbon paste mixture is structurally stable.

Furthermore, using data from the effect of the scan rate, the reaction of gallic acid on the modified electrode is adsorption-controlled. This leads to the possibility of GA being adsorbed to the modified electrode by forming hydrogen bonds with ZrO_2_ nanoparticles on the electrode. With the capability of electron transfer by the ZrO_2_ and graphite (Figure 12), the oxidation of GA induced by the applied voltammetric potential facilitates easy electron transfer. This is in line with the proposal of Chikere et al. (2019) [8], where the authors proposed a mechanism, which suggests that hydrogen bonding was formed between GA and SiO_2_ nanoparticles, with electrons being transferred. However, the GA detection capacity of the ZrO_2_-CPE appears to be about 10-fold better than that of SiO_2_ in graphene, because of the possible stability of the ionic interaction between ZrO_2_ and the graphite.

### 3.8. Effect of Concentration on Gallic Acid Determination.

Differential pulse voltammetry was used to investigate the effect of increasing concentrations of GA on the surface of the amorphous ZrO_2_ nanoparticles modified CPE. The DPV results recorded at a potential range of 0 V to +1.2 V, the scan rate of 100 mVs^−1^, pulse amplitude of 80 mV, and pulse period of 0.2 s were used for the determination of the limit of detection (LOD) of the modified electrode. The increase in GA concentration was studied in the range of 1 × 10^−6^ to 1.0 × 1 × 10^−3^ mol L^−1^. The increasing GA concentration produced a proportional increase in peak current (*I_p_*) on the surface of the ZrO_2_ nanoparticles modified electrode. The measurements produced voltammograms, as can be seen in Figure 13, where the peak currents increased in proportion with the increasing GA concentration. The DPV voltammograms produced two peaks at peak potentials of 0.51 V and 0.85 V as the different gallic acid concentrations were being determined. The peak potential, in this case, shows a negative shift as compared to the peak potential when CV was used. This is attributed to the fact that DPV is more sensitive than CV. This is because charging current (non-faradaic current) is eliminated in DPV measurements, as opposed to CV with the charging and faradaic current [60,61].

The first major peak from the voltammograms was used to produce the analytical calibration graph in Figure 13 (inset). This showed a linear relationship between the peak currents and the increasing concentration of GA, which produced a linear regression equation of *I_p_* = 1125.4 *C* + 2.1113 (*I*_P_: µA, *C*: mmol L^−^^1^ and *R*^2^= 0.9945) within the range of 1 × 10^−6^ to 1.0 × 10^−3^ mol L^−1^. The limit of detection (LOD), which was defined as (3 × *Std*_Blank_)/*m*, where *Std*_Blank_ is the standard deviation of the blank and *m* is the slope; was found to be 1.24 × 10^−^^7^ mol L^−^^1^.

The proposed method was found to be comparable to other methods in the literature, based on the maximum permitted antioxidant levels in food within the EU and North America. The maximum permitted range for GA in the antioxidant level guidelines is 1.2 × 10^−1^ mol L^−1^ to 6.0 mol L^−1^. The recorded LOD being below this stipulated standard permitted level makes the modified electrode suitable for use in GA determination and comparable to others in the literature, as shown in Table 1.

### 3.9. Reproducibility and Repeatability of the Method

The reproducibility of GA determination by the ZrO_2_ nanoparticles modified carbon paste electrode was investigated by using the modified electrodes for the determination of 1 × 10^−3^ mol L^−1^ GA using DPV. Eight independently produced electrodes were used to measure eight different samples of 1 × 10^−3^ mol L^−1^ GA, as can be seen in Figure 14. The oxidation current produced by the eight replicates, independently determined by the different electrodes, gave a relative standard deviation (RSD) of 2.4%, hence showing good reproducibility.

The repeatability of the method was also investigated by using the modified electrodes for eight repeated determination of 1 × 10^−3^ mol L^−1^ GA. The relative standard deviation of the measured peak currents was found to be 3.02%, showing good repeatability of the method.

### 3.10. Stability of the Modified Electrode

The stability of the ZrO_2_ nanoparticles modified CPE was investigated by using three freshly prepared modified CPEs individually to determine GA (1 × 10^−2^ mol L^−1^) using CV, which was then stored properly for 30 days. The electrodes were then further used for the determination of the same concentration of GA. The voltammograms generated from the determination on the 1st and 30th days (Figure 15) demonstrated good stability with a relative standard deviation of 6.2%.

### 3.11. Interference Study on GA Determination

The modified electrode was used to study the interference of various species in the determination of 1 × 10^−3^ mol L^−1^ gallic acid. This was carried out by adding known concentrations of different foreign ions to the analyte and then analysing using DPV. The ionic species used for the studies were Fe^3+^, Ca^2+^, Na^+^, and K^+^; while organic compounds including quercetin, caffeine, caffeic acid, and ascorbic acid were also studied. The use of the different cations was due to their reported complexation with GA in the literature [64,65] and their capability of interfering with the electrochemical determination of gallic acid. The tolerable interference limits of these ions have been defined as the highest concentration or amount that would produce a relative standard deviation of not more than 5%. The results (Figure 16; Table 2) showed relative standard deviations of less than 5%. Therefore, the ions did not interfere with the determination of GA. However, there was some slight positive shift of some of the peak potentials, which still fall within the normal potential range of GA oxidation. Furthermore, the second peak of the GA oxidation showed some minor changes but would not be considered as interference in this case, because the peak considered for electrochemical determination of GA is usually the first peak.

### 3.12. Analytical Application ZrO_2_ Nanoparticles Modified Carbon Paste Electrode

The ZrO_2_ nanoparticles modified carbon paste electrode was used for the determination of GA in red and white wine. Red and white wine were used as samples because of the work done in the literature that suggests the presence of GA in these wines [49,50]. The experiment was carried out with 10 mL of the diluted red and white wine samples, respectively, as a blank. A standard addition of aliquots (1 mL) of a known concentration of GA was made to the wine samples and DPV was used to analyse the samples. The voltammograms (as shown in Figure 17 (red wine) and Table 3) show an oxidation peak, while analysing the blank. Further oxidation peaks were then recorded as the GA concentration in the wine was being increased. Within a concentration range of 0 to 3.3 × 10^−3^ mol L^−1^ the voltammograms were recorded as seen in Figure 17. Increasing the concentration of gallic acid produced a linear regression equation of *I_p_* = 10.9 C + 21.4 and R^2^ = 0.9909.

## 4. Conclusions

Amorphous ZrO_2_ nanoparticles were used for the first time to modify a carbon paste electrode. The electrode was used for the electrochemical determination of gallic acid in a 0.1 mol L^−1^ phosphate buffer at pH 2.0. The amorphous ZrO_2_ nanoparticles modified electrode showed enhanced oxidation peak currents as compared to bare CPE, using cyclic voltammetry and differential pulse voltammetry. The electrode also showed an electrochemical oxidative effect towards the determination of GA within a concentration range of 1 × 10^−^^6^ to 1 × 10^−^^3^ mol L^−^^1^. There was a possible interaction between the ZrO_2_-graphite, which is thought to have influenced the oxidation of GA on the surface of the electrode. The modified electrode demonstrated selectivity towards the determination of GA, as it showed very low interference from other flavonoids like Quercetin and other antioxidants. The electrode was also found to be sensitive, fast, cost-effective, and simple in the determination of gallic acid; it also showed good repeatability, reproducibility, and good long-term stability.

Furthermore, the modified electrode was used for the successful determination of GA in red wine within a concentration range of 0 to 3.3 × 10^−3^ mol L^−1^. The recorded results showed that it was comparable to sensors in the literature based on its LOD, which is far below the maximum permitted GA levels in food samples within the EU and North America.

The use of amorphous ZrO_2_ nanoparticles for the modification of the carbon paste electrode creates a foundation for the subsequent use of amorphous phases of other metal oxide nanoparticles.

## Figures and Tables

**Figure 1 nanomaterials-10-00537-f001:**
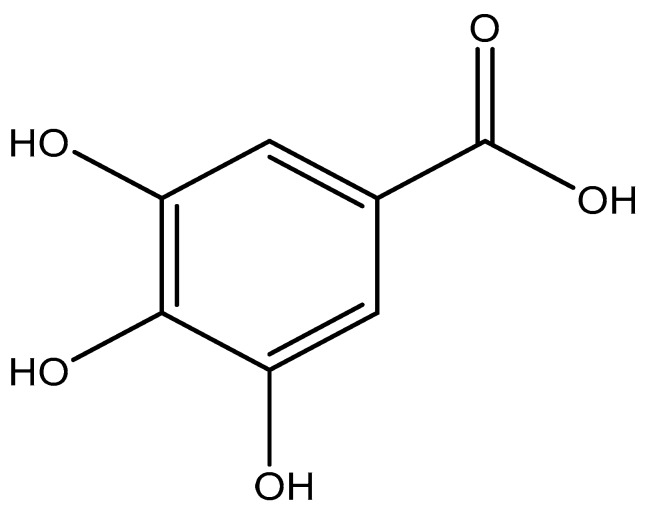
Molecular structure of gallic acid (2,3,4-trihydroxybenzoic acid).

**Figure 2 nanomaterials-10-00537-f002:**
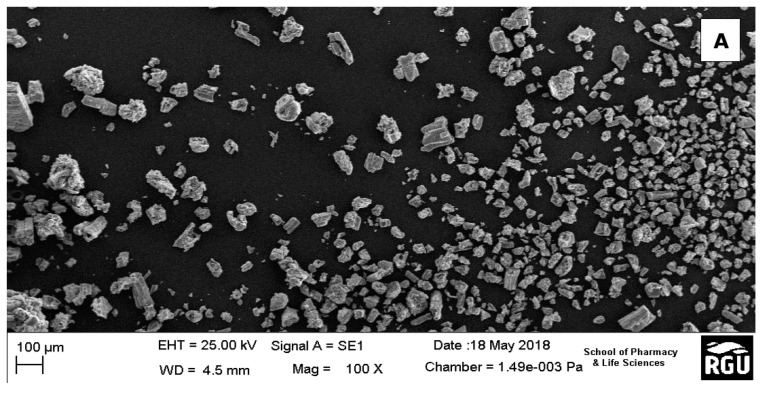
(**a**) SEM image showing the morphology of ZrO_2_ nanoparticles; (**b**) One of the aggregates of the ZrO_2_ nanoparticles.

**Figure 3 nanomaterials-10-00537-f003:**
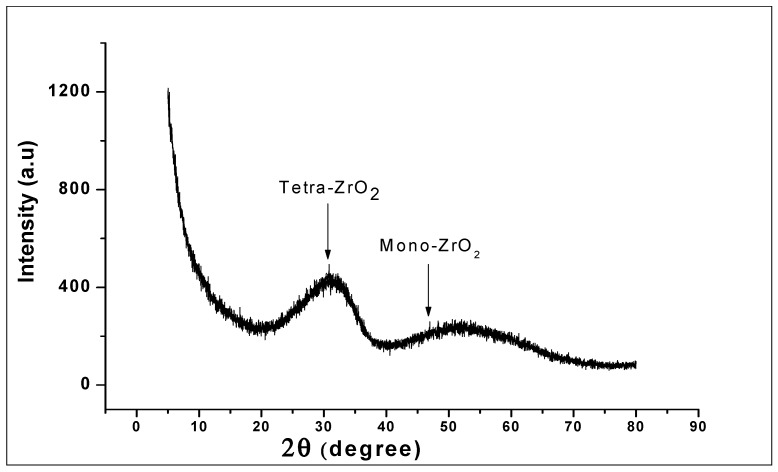
X-Ray Powder Diffraction (XRD) of ZrO_2_ nanoparticles, showing the broad peaks corresponding to tetragonal ZrO_2_ and monoclinic ZrO_2_ structure.

**Figure 4 nanomaterials-10-00537-f004:**
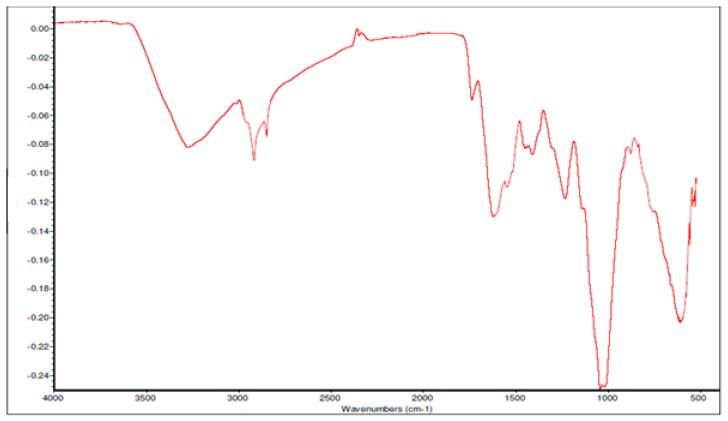
Fourier Transform Infrared spectroscopy (FTIR) spectrum of ZrO_2_ nanoparticles.

**Figure 5 nanomaterials-10-00537-f005:**
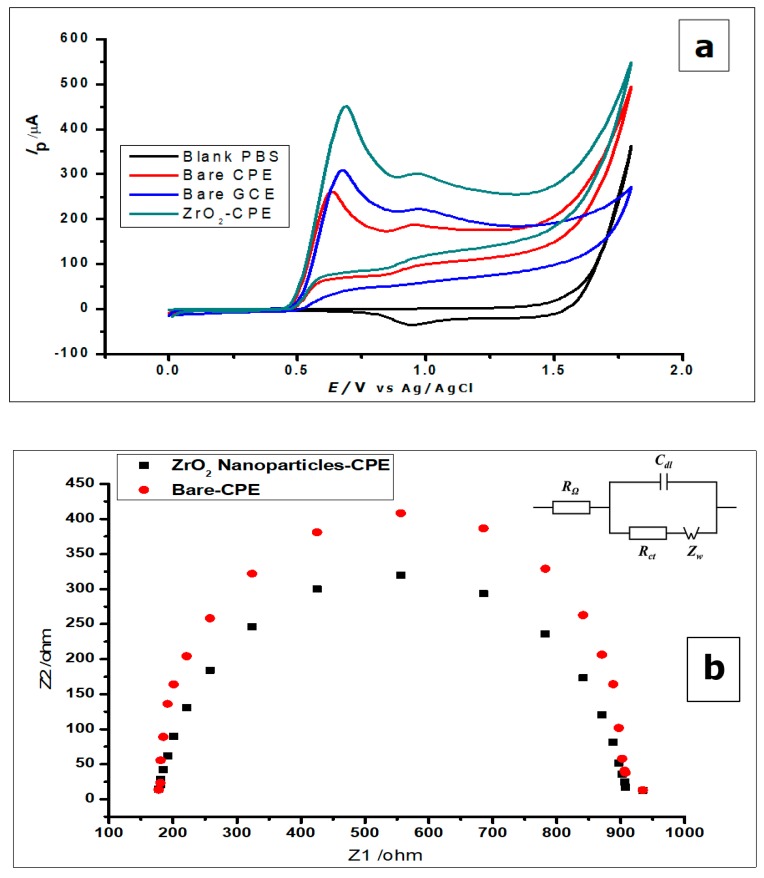
(**a**) Cyclic voltammograms of 1 × 10^−2^ mol L^−1^ GA at ZrO_2_ nanoparticles-modified CPE, bare CPE and bare glassy carbon electrode (GCE) in 0.1 mol L^−^^1^ phosphate buffer of pH 2.0 at a scan rate of 100 mVs^−1^ (**b**) Nyquist plot representing the EIS measurement of 5mmol L^−1^ using the bare CPE and ZrO_2_ nanoparticles (inset) The equivalent circuit showing resistors and capacitor (**c**) The cyclic voltammograms of 1.0 × 10^−3^ mol L^−^ [Fe(CN)_6_]^3−/4−^ redox solution, using bare CPE and ZrO_2_ nanoparticles-modified CPE.

**Figure 6 nanomaterials-10-00537-f006:**
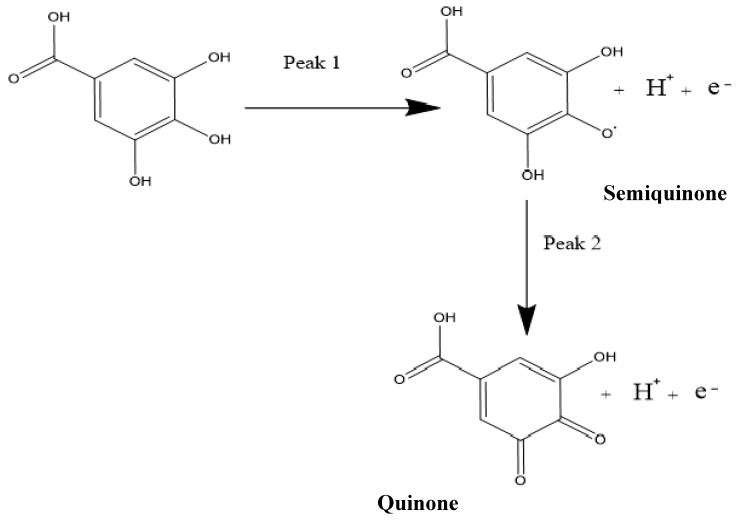
The reaction mechanism of the oxidation of gallic acid, showing the two peaks in cyclic voltammetry (CV) and differential pulse voltammetry (DPV).

**Figure 7 nanomaterials-10-00537-f007:**
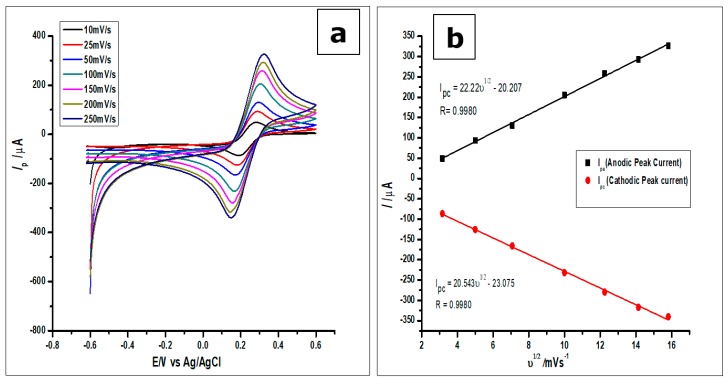
(**a**). Cyclic voltammograms of 1 mmol L^−1^ [Fe(CN)6]^3−/4+^ measured using the ZrO_2_ nanoparticles modified CPE at increasing scan rates from 25–250 mVs^−1^ (**b**). Plots of *I_p_* vs *ν*^1/2^, which would be used to calculate the true surface area.

**Figure 8 nanomaterials-10-00537-f008:**
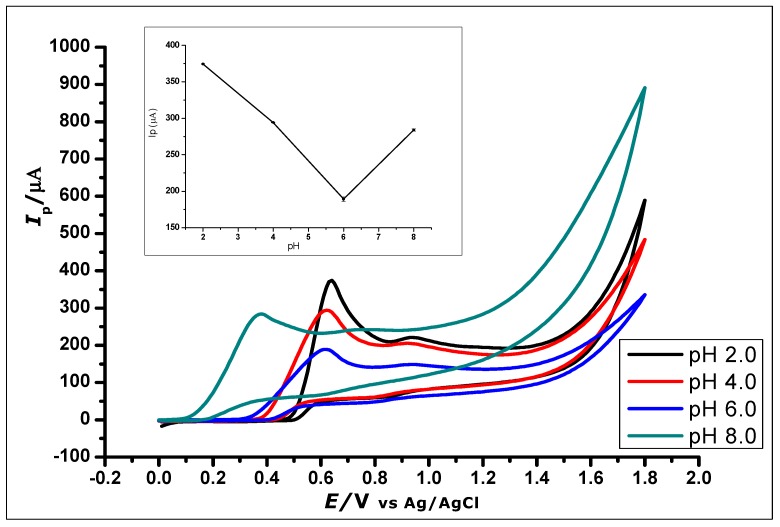
Cyclic Voltammograms showing the effect of pH on the electrochemical behaviour of 1 × 10^−2^ mol L^−1^ Gallic Acid at pH range of 2.0–8.0 (Inset). The plot of the Peak Current (I_p_) against different pH values, showing the effect of pH on the electrochemical behaviour of 1 × 10^−2^ mol L^−1^ gallic acid using ZrO_2_ nanoparticles modified CPE at a scan rate of 100 mVs^−1^.

**Figure 9 nanomaterials-10-00537-f009:**
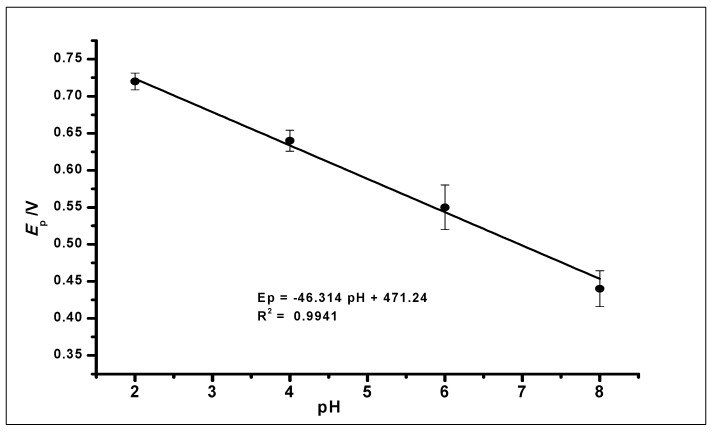
The plot of the peak potential (*E_p_*) against the different pH values of the buffer solution, showing a linear relationship.

**Figure 10 nanomaterials-10-00537-f010:**
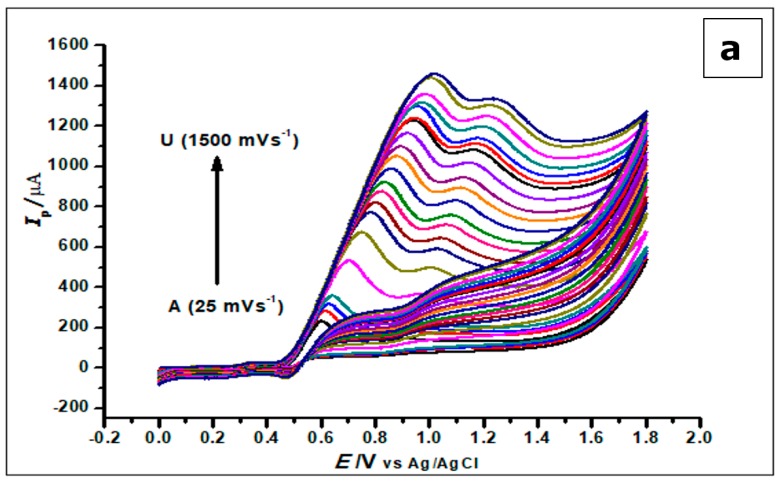
(**a**) Cyclic Voltammograms of 5.0 × 10^−4^ mol L^−1^ gallic acid in phosphate buffer of pH 2.0 determined with a ZrO_2_ nanoparticles-modified CPE at different scan rates from 25 to 1500 mVs^−1^. (**b**). The plot of the scan rate vs. the peak current (**c**). The plots of log *I*_p_ vs. log *ν* showing straight lines.

**Figure 11 nanomaterials-10-00537-f011:**
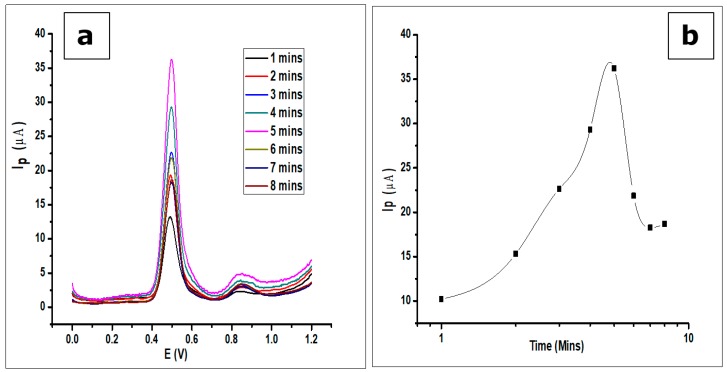
(**a**) DPV voltammograms showing the effect of adsorption time on the determination of 1 × 10^−3^ mol L^−1^ GA in a 0.1 mol L^−1^ phosphate buffer of pH 2.0 at a scan rate of 100 mVs^−1^ (**b**). The plots showing the same effect of adsorption time.

**Figure 12 nanomaterials-10-00537-f012:**
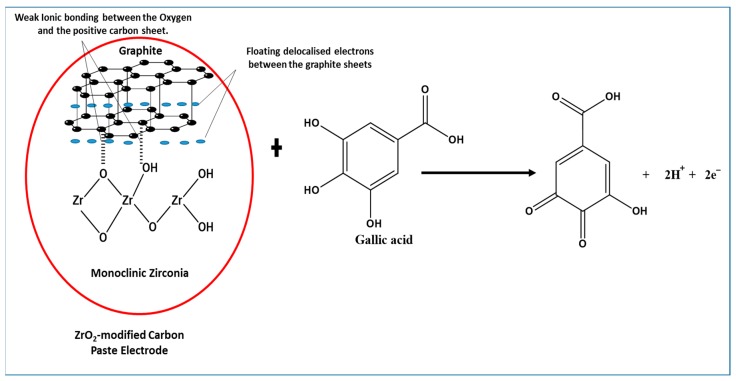
Proposed interaction between ZrO_2_ nanoparticles (monoclinic zirconia) and graphite, showing a weak ionic bond between some of the oxygen and the positive graphite sheet, towards electrochemical determination of gallic acid.

**Figure 13 nanomaterials-10-00537-f013:**
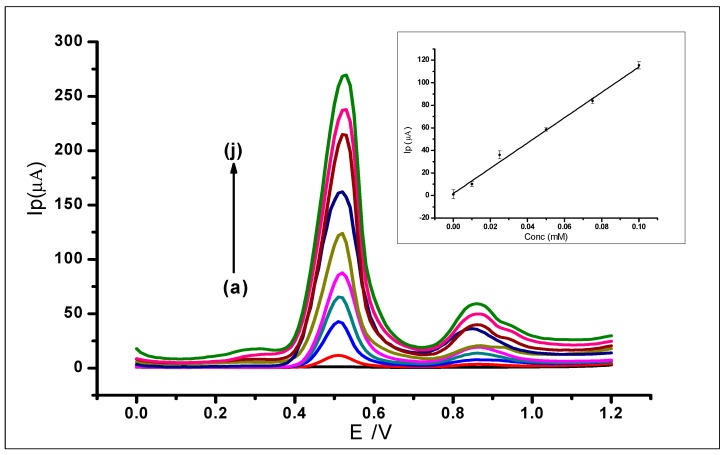
Differential voltammograms of various concentrations of GA at ZrO_2_ nanoparticles modified CPE in a 0.1 mol L^−1^ phosphate buffer at 2.0, scan rate of 100 mVs^−1^, with voltammograms (a–j) that corresponds to the following concentrations: (a) Blank PBS, (b) 0.01 mmol L^−1^, (c) 0.025 mmol L^−1^, (d) 0.05 mmol L^−1^, (e) 0.075 mmol L^−1^, (f) 0.1 mmol L^−1^, (g) 0.25 mmol L^−1^, (h) 0.5 mmol L^−1^, (i) 0.075 mmol L^−1^ (j) 1 mmol L^−1^.

**Figure 14 nanomaterials-10-00537-f014:**
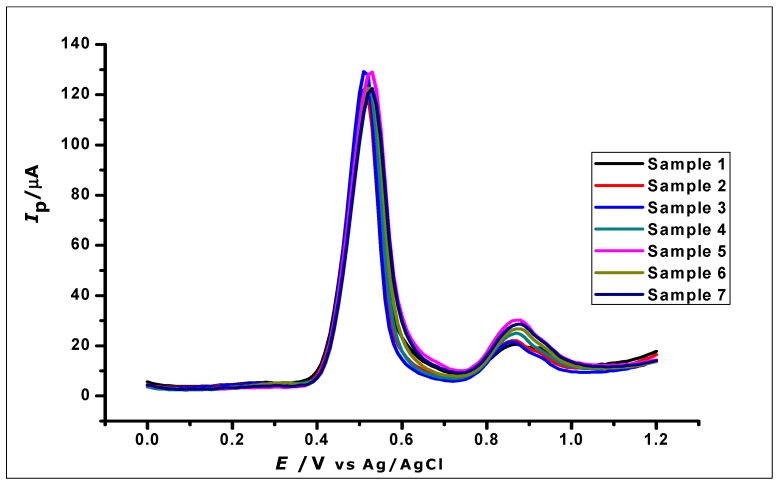
DPV Voltammogram of 1 × 10^−3^ mol L^−1^ gallic acid using ZrO_2_ nanoparticles modified-CPE to demonstrate reproducibility.

**Figure 15 nanomaterials-10-00537-f015:**
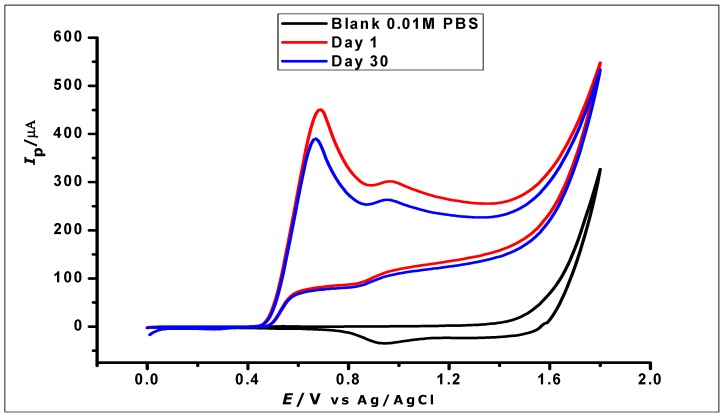
Cyclic voltammograms showing the determination of 1 × 10^−2^ mol L^−1^ GA in 1 × 10^−2^ phosphate buffer at pH 2.0 and scan rate of 100 mVs^−1^ on a day one and day thirty.

**Figure 16 nanomaterials-10-00537-f016:**
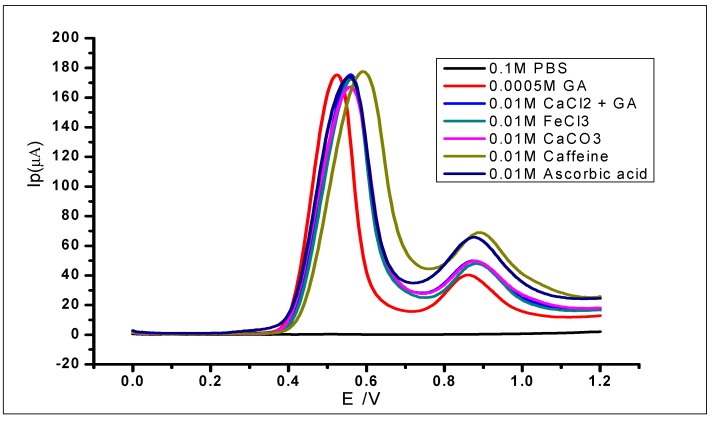
DPV voltammograms of 5 × 10^−4^ mol L^−1^ gallic acid with different concentrations of known.

**Figure 17 nanomaterials-10-00537-f017:**
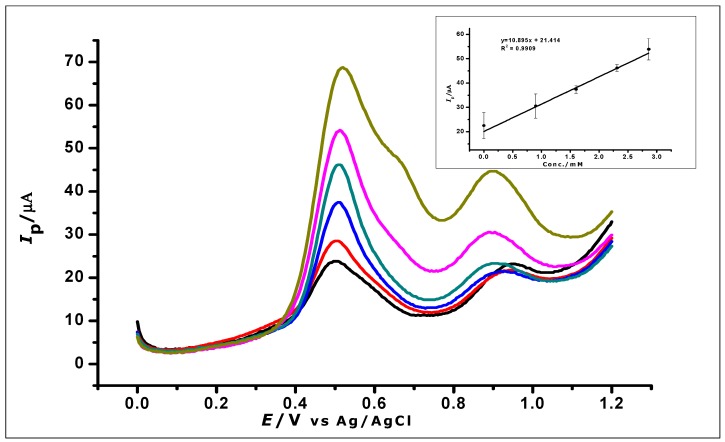
DPV voltammograms show the determination of gallic acid in a real sample of red wines; (Inset) calibration graph created using the analysis of the real sample.

**Table 1 nanomaterials-10-00537-t001:** Electrochemical methods used in the determination of gallic acid and their efficiency.

Electrode	Method	Medium Analysed	Linear Range (mol L^−^^1^)	LOD/(mol L^−^^1^)	Reference
Graphite electrode modified with thionine (TH) and Nickel hexacyanoferrate (NiHCF)	CV, DPV	Green Tea Milk	4.99 × 10^−^^6^ to 1.2 x 10^−^^3^	1.66 × 10^−^^6^	[62]
Carbon paste electrode modified with carbon nanotubes	CV, DPV	Red Wine White Wine	5 × 10^−^^7^ to 1.5 x 10^−^^5^	3.0 × 10^−^^7^	[49]
Carbon paste electrode modified with TiO_2_ nanoparticles	DPV	Tap Water Green Tea Black Tea	2.5 × 10^−^^6^ to 1.5 x 10^−^^4^	9.4 × 10^−^^7^	[6]
Carbon paste electrode modified with SiO_2_ nanoparticles	DPV	Tap Water Orange Juice Green Tea Black Tea	8.0 × 10^−^^7^ to 1.0 × 10^−^^4^	2.5 × 10^−^^7^	[7]
Glassy carbon electrode modified with the electrodeposition of Zn-Al-NO_3_ layered double hydroxide film.	DPV	Green Tea	4.0 × 10^−^^6^ to 6 × 10^−^^4^	1.6 × 10^−^^6^	[63]
Glassy Carbon Electrode modified with polyethyleneimine-functionalized graphene oxide (PEI-rGO)	CV	Green Tea Black Tea	5.8 × 10^−^^7^ to 5.8 × 10^−^^4^	4.1 × 10^−^^7^	[48]
Glassy carbon electrode modified with SiO_2_ nanoparticles and graphene oxide nanocolloids	CV, DPV	Red Wine White Wine Orange Juice	6.3 × 10^−^^6^ to 1.0 × 10^−^^3^	2.09 × 10^−^^6^	[8]
ZrO_2_ nanoparticles modified carbon paste electrode	CV, DPV	Red Wine White Wine	1 × 10^−^^6^ to 1.0 × 10^−^^3^	1.24 × 10^−^^7^	This work

**Table 2 nanomaterials-10-00537-t002:** Effects of various substances on the determination of 1 × 10^−3^ mol L^−1^ gallic acid.

Interfering Species	Amount in Solution (mol L^-1^)	Relative Standard Deviation (%)
Ca^2+^	1 × 10^−1^	±3.70
Na^+^	1 × 10^−1^	±1.29
Fe^3+^	1 × 10^−1^	±1.93
Cl^−^	1 × 10^−1^	±2.01
CO_3_^2−^	1 × 10^−1^	±1.45
Ascorbic acid	1 × 10^−3^	±0.15
Caffeic acid	1 × 10^−3^	±2.80
Caffeine	1 × 10^−3^	±3.89
Quercetin	1 × 10^−3^	±4.64

**Table 3 nanomaterials-10-00537-t003:** Results of the analysis of GA in the spiked red and white wine.

Samples	Added (mmol L^−1^)	Found (mmol L^−1^)	Recovery (%)	Relative Error
Red Wine	0	0.1030	-	-
	0.9	0.8360	92.8	±7.2
	1.6	1.469	91.8	±8.2
	2.3	2.273	98.7	±1.3
	2.86	2.98	104.2	±4.2
White Wine	0	0.049	-	-
	0.9	0.95	105.5	±5.5
	1.6	1.66	103.75	±3.75
	2.3	2.24	97.39	±2.61

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
