# Peer review of "Interaction between Amorphous Zirconia Nanoparticles and Graphite: Electrochemical Applications for Gallic Acid Sensing Using Carbon Paste Electrodes in Wine"

_nanomaterials, 2020, doi:10.3390/nano10030537_

Round 1

Reviewer 1 Report

The authors report the use of ZrO2 nanoparticles to improve the electrochemical detection of gallic acid at carbon paste electrodes. The response of ZrO2 modified electrodes was characterized by cyclic voltammetry and differential pulse voltammetry and compared with carbon paste electrodes without ZrO2 and with glassy carbon electrodes. The modified electrodes were further used for the quantification of gallic acid in wine samples.

The authors have addressed some of the issues raised on the 1st submission which has helped improving the manuscript. Unfortunately, language and sentence construction was not revised, so the paper remains very difficult to read (e.g. page 6 lines 191-192 and 207-208, page 12 lines 306-308 and 315-317, page 15 lines 388-389, 393-396, page 18 lines 460-462). Besides, the manuscript fails to provide a critical discussion on several key aspects of the results.

The authors have not commented on the reasons for the aggregation of the ZrO2 samples or on the discrepancy between sizing results obtained in SEM and dynamic light scattering measurements. If micro-particles are present, this should be considered when discussing the reasons for the enhanced current response to gallic acid.

According to new results about the areas of the control electrodes, there is a significant increase of electroactive area of the carbon paste electrodes in the presence of ZrO2 (ca. 8 times), this should be highlighted in the discussion.

The authors cannot compare the electrochemical response obtained in the absence and in the presence of gallic acid in solution to conclude the ZrO2 contribute to the enhancement of oxidation currents.

No valid explanation is given for the decrease of oxidation currents after 5 minutes of adsorption of gallic acid, if there is saturation of the electrode surface by gallic acid, as mentioned in the paper, shouldn’t the currents stabilize?

Specific points, such as the scan rate discrepancy between experimental section and the results and discussion, were not taken into consideration.

The FTIR spectrum represented in Figure 4 appears to have been acquired in transmission not absorption mode as mentioned in the experimental section.

The highlights do not reveal the novelty or impact of the work and the graphical abstract shows DPV measurements not cyclic voltammograms.

Author Response

Revision of Manuscript for Publication

General Remarks

Thank you very much, once again for the opportunity to revise our manuscript. The revised version of the manuscript (Manuscript ID: nanomaterials-676979) titled “Interaction between amorphous zirconia nanoparticles and graphite: Electrochemical applications for gallic acid sensing using carbon paste electrodes in wine” has now been submitted with the relevant areas updated as appropriate.

We appreciate all the thoughtful comments and recommendations. The manuscript has been carefully looked through and revised in line with the comments put forward by the reviewers. All the points raised by the reviewers have been carefully considered. The revision, changes and explanations on the manuscript are as follows:-

Response to Reviewer One

General and specific remarks

The authors report the use of ZrO2 nanoparticles to improve the electrochemical detection of gallic acid at carbon paste electrodes. The response of ZrO2 modified electrodes was characterized by cyclic voltammetry and differential pulse voltammetry and compared with carbon paste electrodes without ZrO2 and with glassy carbon electrodes. The modified electrodes were further used for the quantification of gallic acid in wine samples.

The authors have addressed some of the issues raised on the 1st submission which has helped improving the manuscript. Unfortunately, language and sentence construction were not revised, so the paper remains very difficult to read (e.g. page 6 lines 191-192 and 207-208, page 12 lines 306-308 and 315-317, page 15 lines 388-389, 393-396, page 18 lines 460-462). Besides, the manuscript fails to provide a critical discussion on several key aspects of the results.

The authors have not commented on the reasons for the aggregation of the ZrO2 samples or on the discrepancy between sizing results obtained in SEM and dynamic light scattering measurements. If micro-particles are present, this should be considered when discussing the reasons for the enhanced current response to gallic acid.

According to new results about the areas of the control electrodes, there is a significant increase of electroactive area of the carbon paste electrodes in the presence of ZrO2 (ca. 8 times), this should be highlighted in the discussion.

The authors cannot compare the electrochemical response obtained in the absence and in the presence of gallic acid in solution to conclude the ZrO2 contribute to the enhancement of oxidation currents.

No valid explanation is given for the decrease of oxidation currents after 5 minutes of adsorption of gallic acid, if there is saturation of the electrode surface by gallic acid, as mentioned in the paper, shouldn’t the currents stabilize?

Specific points, such as the scan rate discrepancy between experimental section and the results and discussion, were not taken into consideration.

The FTIR spectrum represented in Figure 4 appears to have been acquired in transmission not absorption mode as mentioned in the experimental section.

The highlights do not reveal the novelty or impact of the work and the graphical abstract shows DPV measurements not cyclic voltammograms.

Response: We are very grateful that the reviewers have thoroughly gone through the manuscript, with the very constructive and feedbacks and comments. Following the valuable advice received, we have carefully addressed each one of the comments. We thank the reviewers once again for their well thought through comments

Comment 1: Unfortunately, language and sentence construction were not revised, so the paper remains very difficult to read (e.g. page 6 lines 191-192 and 207-208, page 12 lines 306-308 and 315-317, page 15 lines 388-389, 393-396, page 18 lines 460-462). Besides, the manuscript fails to provide a critical discussion on several key aspects of the results.

Response: We thank the reviewers for pointing out these language drawbacks. We appreciate them and have made the corrections as follows: -

Lines 192-195 which is now lines 192-194 has been revised to, “The influence of the accumulation time of GA (1 x 10-3 mol L-1), on the surface of the modified electrode, was studied. The accumulation time is an important parameter that affects the amount of GA adsorbed on the electrode surface. The voltammograms for the GA determination whilst increasing the accumulation times every minute up to eight minutes (0-480 sec), were recorded.”

Lines 207-208 which is now lines 210-212 have been revised and written; thus, “Then aliquots (1 mL) of GA standard (1 x 10-2 mol L-1) were added into the wine samples and vigorously stirred with a magnetic stirrer for two mins. After every addition, the differential pulse voltammogram measurements were recorded”.

Lines 306-308 which is now lines 315-317 have been revised thus, “The EIS was used to probe the difference in the behaviour of the bare CPE and the ZrO2 nanoparticles modified CPE. Both electrodes were used to measure [Fe(CN)6]3-/4- (5 mmol L-1) redox solution, as can be seen by the Nyquist plots in Figure 5b”.

Lines 315-317 which is now lines 324-326 have been revised thus, “Furthermore, the difference in electrochemical capacity of the two electrodes were further investigated, by using the electrodes to measure the redox couple [Fe(CN)6]3-/4- (1.0 x 10-3 mol L-1), whilst using CV at a scan rate of 100 mVs-1”.

Lines 388-389 which is now lines 397-400 have been revised to the following:- “The increase in scan rate produces a positive shift in the peak potential and a linear increase in peak currents. This phenomenon suggests a kinetic limitation in reaction of GA at the surface of the modified electrode [52]”.

Lines 393-396 which is now lines 403-407, have been revised and updated to the following:- “Plotting the relationship between the logarithm of peak current ( log Ip) and the logarithm of the scan rate (log v) in 25-1000 mVs-1 scan range, also generated a straight line with a linear regression equation: - Log Ip (µA) = 1.7884 Log v (mV/s) + 2.7607 and R2 = 0.9902. These results indicate a diffusion and an adsorption-controlled reaction of GA at the ZrO2 nanoparticles modified CPE as can be seen in Figure 10b”.

Lines 460-462 which is now lines 474-476 have been revised to the following:- “This is attributed to the fact that DPV is more sensitive than CV. Hence, there is no charging current (non-faradaic current) needed for DPV as opposed to CV with the charging and the faradaic current [60,61]”.

Comments 2: The authors have not commented on the reasons for the aggregation of the ZrO2 samples or on the discrepancy between sizing results obtained in SEM and dynamic light scattering measurements. If micro-particles are present, this should be considered when discussing the reasons for the enhanced current response to gallic acid.

Response: Thank you for the kind comments. We have looked at the ZrO2 nanoparticles and generally, when ZrO2 nanoparticles are not annealed they tend to form aggregates, with non-homogenous size distributions. With increasing annealing temperatures between 500 °C to 800 °C, ZrO2 nanoparticles show monoclinic phases. Meanwhile, in our work, the XRD results show mainly the tetragonal phases of the compound. The electrochemical modification and analysis were carried out as amorphous ZrO2 nanoparticles modification of CPE. Thus, the manuscript has been updated at lines 224-228 as follows: - “This agrees with the work of Opalinska et. al [39] where they found out that larger particle sizes of ZrO2 nanoparticles, formed aggregates because of the particle sintering. They also described the specific surface area of the particles depends on the exposed surface area of the aggregated. However, with increasing annealing temperatures of between 500 °C to 800°C, the nanoparticles become independent particles in the monoclinic phase”.

Comments 3: According to new results about the areas of the control electrodes, there is a significant increase of electroactive area of the carbon paste electrodes in the presence of ZrO2 (ca. 8 times), this should be highlighted in the discussion.

The authors cannot compare the electrochemical response obtained in the absence and in the presence of gallic acid in solution to conclude the ZrO2 contribute to the enhancement of oxidation currents.

Response: We are very grateful for the kind review and the thoughtful comment. The discussion section has been updated and with additional information to highlight the effect of the increased electroactive surface area. This can be seen in Lines 270-272 as follows: - “The amorphous ZrO2 nanoparticles used for the modification of CPE, increased the electroactive surface area of the electrode about eight fold; from 0.0345 cm2 to 0.276 cm2”.

Regarding the comparison of the electrochemical response obtained in the absence and presence of GA in solution to conclude that, enhancement was because of ZrO2. We have revised and updated the section to read as follows in Lines 301-304: - “Furthermore, at the same potential range of 0.0 – 1.8 V and over the range where GA oxidation peak was observed, no oxidation peak was present with only buffer (0.1 mol L-1 phosphate), when all the electrodes were used. However, when GA was added to the phosphate buffer solution, all the electrodes produced the anodic peaks, with the ZrO2 nanoparticles showing the highest peak current”.

Comment 4: No valid explanation is given for the decrease of oxidation currents after 5 minutes of adsorption of gallic acid, if there is saturation of the electrode surface by gallic acid, as mentioned in the paper, shouldn’t the currents stabilize?

Response: Thank you for this comment. The interaction of GA with the surface of the modified electrode as described is an adsorption-controlled reaction. The adsorption of GA on the surface of the electrode, does not only generate peak currents, but commonly develop electrode fouling, which disrupt further electron transfer. Hence, we have revised the explanation to read as follows from Lines 416-419: - “Which means, the electrode at this point does no longer transfer electrons as much as it did in the beginning, hence the decrease in peak current. This is as a result of electrode fouling which is common with the production of phenoxy radicals and their subsequent reaction with phenol creating a polymeric adherent film that is deposited on the electrode surface [53]”.

Comment 5: Specific points, such as the scan rate discrepancy between experimental section and the results and discussion, were not taken into consideration.

Response: We are very grateful that reviewers pointed this out. We have checked the manuscript and can confirm that, the said discrepancy may be due to the increasing in scan rate range from 25 – 250 mVs-1 used when measuring [Fe(CN)6]3-/4+ for electroactive surface area of the electrode. The scan rate range of 25 – 1500 mVs-1 was used to measure GA to determine if the reaction, was an adsorption controlled reaction and/or diffusion controlled at the surface of the electrode. With the specific range of 25-1000 mVs-1 that produced the linear relationships.

Hence, the following highlights have been made on the transcript.

Lines 330-331: “A concentration of 1.0 mmol L-1 K3[Fe(CN)6]3-/4+ dissolved in 0.1 mol L-1 KCl was measured using CV and the results recorded at increasing scan rates from 25 mVs-1 to 250 mVs-1

Lines 388-390: “The effect of scan rate on the electrochemical behaviour of 1 x 10-3 mol L-1 GA at the ZrO2 nanoparticles CPE at different scan rates ranging from 25 mVs-1 – 1500 mVs-1 was investigated using cyclic voltammetry”.

Line 394: 25 mVs-1 to 250 mVs-1

Lines 403-407: “Plotting the peak current (Ip) against the scan rate (v) in the scan rate range of 25 – 1000 mVs-1 produced a straight line, suggesting that the GA interacts with the ZrO2 nanoparticles”

Comment 6: The FTIR spectrum represented in Figure 4 appears to have been acquired in transmission not absorption mode as mentioned in the experimental section.

Response: We thank you very much for this observation. The FTIR images shown in Figure 4 were rightly acquired in Transmission mode as the reviewers noted and the experimental section has been revised to transmittance in Line 156 page 5.

Comments 7: The highlights do not reveal the novelty or impact of the work and the graphical abstract shows DPV measurements not cyclic voltammograms.

Response: The authors are very grateful that the reviewers pointed these out, with respect to the highlights. We have looked at the highlights and graphical abstract and revised them as appropriate.

Lines 12-16:   

  • For the first time synthesised amorphous zirconium dioxide nanoparticles modified CPE for the electrochemical determination of gallic acid
  • An interaction of the amorphous ZrO2 nanoparticles with the graphite has been proposed
  • Amorphous ZrO2 nanoparticles modified carbon paste electrode has been used for the detection of gallic acid in Wine samples.

Reviewer 2 Report

The revision addressed reviewers' questions.

Author Response

Dear Reviewer,

Many thanks for your positive comments

Round 2

Reviewer 1 Report

The authors addressed several issues in the revised version of the manuscript. However, some points still require further attention. Importantly, the manuscript would benefit from extensive English revision.

Point 1) Rephrasing

It is not clear what the authors mean by electrochemical capacity in line 325. The sentence appears to refer to the comparison of the electrochemical response of the ZrO2 modified and the bare CPE.

Please rephrase lines 477-478; it is not a matter of needing charging currents, they can be eliminated in DPV measurements but are present in CV.

Point 2) ZrO2 particle size

The authors have not commented on the different sizes obtained for the ZrO2 particles from SEM and dynamic light scattering measurements.

Furthermore, in the paper by Opalinska et al. sintering explains the higher apparent particle size observed with increasing annealing temperatures, from 400 to 1100 ºC, which is not the case in the current manuscript, as much lower temperatures were used.

The results show that micro sized ZrO2 aggregates were synthesized in the current work. The reasons for the aggregation may be out of the scope of the present manuscript, still the fact that they are micro-particles should be clearly stated in the discussion.

Point 3) Measurements in buffer (lines 302-307)

It is impossible to determine any effects of the ZrO2 particles on gallic acid oxidation when performing measurements in buffer alone. Such control assays demonstrate that there is no current response from the buffer system or electrodes in the potential range at which gallic acid oxidation is measured. Therefore, it is not clear why the authors use these control assays to confirm the effect of ZrO2 particles in gallic acid electrochemical oxidation. This can only be done when gallic acid is present in solution and by comparison with the electrodes that do not contain ZrO2. In fact, the enhancement of gallic acid oxidation current in the presence of ZrO2 is appropriately reported on lines 263-265 and again discussed on lines 278-283.

Point 4) Adsorption of gallic acid on the electrode surface.

Electrode fouling is an acceptable explanation for the decreased oxidation currents with gallic acid adsorption time (lines 420-422). Unfortunately, the previous sentence (419-420) is redundant and should be removed or revised; if the current decreases there is no need to explain that less electrons are transferred.

Point 5) Scan rate discrepancy

As requested, the scan rate originally indicated in the experimental section (10 mV.s-1) has been corrected (line 184) to match the remaining of the manuscript (100 mV.s-1). As for the other changes introduced in the revised version, although potentially useful, the authors should consider rephrasing lines 330-331 to clarify that [Fe(CN)6]3-/4+ electrochemical response was measured using CV at different scan rates, not the concentration as the sentence implies. Also, note that K3[Fe(CN)6]3-/4+ is not the correct notation.

Point 7) Highlights

Abbreviation of CPE is not used consistently in the highlights. The first highlight should also be revised, as it appears a verb is missing in the sentence, which is otherwise difficult to understand.

Author Response

Revision of Manuscript for Publication

General Remarks

Thank you very much, once again for the opportunity to revise our manuscript. The revised version of the manuscript (Manuscript ID: nanomaterials-676979) titled “Interaction between amorphous zirconia nanoparticles and graphite: Electrochemical applications for gallic acid sensing using carbon paste electrodes in wine” has now been revised and updated. The revised copy has been re-submitted.

We are very appreciative of the thoughtful comments and recommendations. In the manuscript we have responded to the comments and recommendations appropriately. The revision, changes and explanations on the manuscript are as follows:-

Response to Reviewer One

General and specific remarks

The authors addressed several issues in the revised version of the manuscript. However, some points still require further attention. Importantly, the manuscript would benefit from extensive English revision.

Point 1) Rephrasing

It is not clear what the authors mean by electrochemical capacity in line 325. The sentence appears to refer to the comparison of the electrochemical response of the ZrO2 modified and the bare CPE.

Please rephrase lines 477-478; it is not a matter of needing charging currents, they can be eliminated in DPV measurements but are present in CV.

Point 2) ZrO2 particle size

The authors have not commented on the different sizes obtained for the ZrO2 particles from SEM and dynamic light scattering measurements.

Furthermore, in the paper by Opalinska et al. sintering explains the higher apparent particle size observed with increasing annealing temperatures, from 400 to 1100 ºC, which is not the case in the current manuscript, as much lower temperatures were used.

The results show that micro sized ZrO2 aggregates were synthesized in the current work. The reasons for the aggregation may be out of the scope of the present manuscript, still the fact that they are micro-particles should be clearly stated in the discussion.

Point 3) Measurements in buffer (lines 302-307)

It is impossible to determine any effects of the ZrO2 particles on gallic acid oxidation when performing measurements in buffer alone. Such control assays demonstrate that there is no current response from the buffer system or electrodes in the potential range at which gallic acid oxidation is measured. Therefore, it is not clear why the authors use these control assays to confirm the effect of ZrO2 particles in gallic acid electrochemical oxidation. This can only be done when gallic acid is present in solution and by comparison with the electrodes that do not contain ZrO2. In fact, the enhancement of gallic acid oxidation current in the presence of ZrO2 is appropriately reported on lines 263-265 and again discussed on lines 278-283.

Point 4) Adsorption of gallic acid on the electrode surface.

Electrode fouling is an acceptable explanation for the decreased oxidation currents with gallic acid adsorption time (lines 420-422). Unfortunately, the previous sentence (419-420) is redundant and should be removed or revised; if the current decreases there is no need to explain that less electrons are transferred.

Point 5) Scan rate discrepancy

As requested, the scan rate originally indicated in the experimental section (10 mV.s-1) has been corrected (line 184) to match the remaining of the manuscript (100 mVs-1). As for the other changes introduced in the revised version, although potentially useful, the authors should consider rephrasing lines 330-331 to clarify that [Fe(CN)6]3-/4+ electrochemical response was measured using CV at different scan rates, not the concentration as the sentence implies. Also, note that K3[Fe(CN)6]3-/4+ is not the correct notation.

Point 7) Highlights

Abbreviation of CPE is not used consistently in the highlights. The first highlight should also be revised, as it appears a verb is missing in the sentence, which is otherwise difficult to understand.

Response: We are very grateful for the meticulous review of the manuscript, with the constructive feedbacks, comments and recommendations. Following the valuable advice received, we have carefully addressed each one of the comments. We thank the reviewers once again for their well thought through comments.

Comment 1: The authors addressed several issues in the revised version of the manuscript. However, some points still require further attention. Importantly, the manuscript would benefit from extensive English revision.

Response: We thank the reviewers so much for the comment. We have taken that on board and made appropriate amendments.

Comment 2: Point 1) Rephrasing

It is not clear what the authors mean by electrochemical capacity in line 325. The sentence appears to refer to the comparison of the electrochemical response of the ZrO2 modified and the bare CPE.

Please rephrase lines 477-478; it is not a matter of needing charging currents, they can be eliminated in DPV measurements but are present in CV.

Response: We thank the reviewers for pointing this out. The phrase has been changed appropriately in Lines 324-325 thus, “Furthermore, the difference in electrochemical response of the ZrO2 nanoparticles modified CPE and the bare CPE were further investigated”.

Lines 477-478 which is now lines 475-477 has been revised to the following, “This is because charging current (non-faradaic current) is eliminated in DPV measurements as opposed to CV with the charging and the faradaic current [60,61]”.

Comment 3: Point 2) ZrO2 particle size

The authors have not commented on the different sizes obtained for the ZrO2 particles from SEM and dynamic light scattering measurements.

Furthermore, in the paper by Opalinska et al. sintering explains the higher apparent particle size observed with increasing annealing temperatures, from 400 to 1100 ºC, which is not the case in the current manuscript, as much lower temperatures were used.

The results show that micro sized ZrO2 aggregates were synthesized in the current work. The reasons for the aggregation may be out of the scope of the present manuscript, still the fact that they are micro-particles should be clearly stated in the discussion.

Response: We are very grateful for the kind review and the thoughtful comment. Like it was earlier explained, ZrO2 nanoparticles are not annealed they tend to form aggregates, with non-homogenous size distributions. With increasing annealing temperatures between 500 °C to 800 °C, ZrO2 nanoparticles show monoclinic phases. However, in our work, the XRD results which is consistent with amorphous ZrO2 compounds show predominantly tetragonal phases. Thus, the manuscript has been further updated thus in lines 224-228, “These micro-particle sizes of some of the ZrO2 are consistent with the XRD results, which shows the compounds are in amorphous form. This agrees with the work of Opalinska et. al [39] where they found out that the ZrO2 nanoparicles form aggregates at temperatures lower than the annealing temperatures of between 500°C to 1100°C. However, with increasing annealing temperatures the nanoparticles become independent particles in the monoclinic phase.”

Comment 4: Point 3) Measurements in buffer (lines 302-307)

It is impossible to determine any effects of the ZrO2 particles on gallic acid oxidation when performing measurements in buffer alone. Such control assays demonstrate that there is no current response from the buffer system or electrodes in the potential range at which gallic acid oxidation is measured. Therefore, it is not clear why the authors use these control assays to confirm the effect of ZrO2 particles in gallic acid electrochemical oxidation. This can only be done when gallic acid is present in solution and by comparison with the electrodes that do not contain ZrO2. In fact, the enhancement of gallic acid oxidation current in the presence of ZrO2 is appropriately reported on lines 263-265 and again discussed on lines 278-283.

Response: We thank you very much for this observation. The measurement of the blank buffer was to confirm that, there was no current response, when the modified and the bare CPE were used. Hence that section of the manuscript, has been edited in lines 300-304, “Furthermore, the both electrodes were used to measure the blank buffer (0.1 mol L-1 phosphate), to confirm the buffer showed no current response at the same potential range of 0.0 – 1.8 V (Figure 5a). No oxidation peak was present with only the buffer (0.1 mol L-1 phosphate), when all the electrodes were used. However, when GA was added to the phosphate buffer solution, all the electrodes produced the anodic peaks, with the ZrO2 nanoparticles showing the highest peak current.” 

Comment 5: Point 4) Adsorption of gallic acid on the electrode surface.

Electrode fouling is an acceptable explanation for the decreased oxidation currents with gallic acid adsorption time (lines 420-422). Unfortunately, the previous sentence (419-420) is redundant and should be removed or revised; if the current decreases there is no need to explain that less electrons are transferred.

Response: Thank you for the observation. The sentence cited has been removed and the section from lines 415-418 reads thus, “Thus, any further scan does not produce an increase in peak current but a decrease. This is as a result of electrode fouling which is common with the production of phenoxy radicals and their subsequent reaction with phenol creating a polymeric adherent film that is deposited on the electrode surface [53]. This agrees with the work done by Tashkhourian [7] and Chikere et al. [8].”

Comment 6: Point 5) Scan rate discrepancy

As requested, the scan rate originally indicated in the experimental section (10 mV.s-1) has been corrected (line 184) to match the remaining of the manuscript (100 mVs-1). As for the other changes introduced in the revised version, although potentially useful, the authors should consider rephrasing lines 330-331 to clarify that [Fe(CN)6]3-/4+ electrochemical response was measured using CV at different scan rates, not the concentration as the sentence implies. Also, note that K3[Fe(CN)6]3-/4+ is not the correct notation.

Response: We are grateful for this comment. The section has been rephrased and presented thus in lines 329-330, “The redox solution [Fe(CN)6] (1.0 x 10-3 mol L-1) was measured using CV and the results recorded at different scan rates from 25 mVs-1 to 250 mVs-1”.

Comment 7: Point 7) Highlights

Abbreviation of CPE is not used consistently in the highlights. The first highlight should also be revised, as it appears a verb is missing in the sentence, which is otherwise difficult to understand.

Response: We thank the reviewers for pointing this out. The highlights have been appropriately revised thus,

  • For the first time, synthesised amorphous zirconium dioxide nanoparticles modified carbon paste electrode (CPE), has been used for the electrochemical determination of gallic acid
  • An interaction of the amorphous ZrO2 nanoparticles with the graphite has been proposed
  • Amorphous ZrO2 nanoparticles modified CPE has been used for the detection of gallic acid in Wine samples.

Round 3

Reviewer 1 Report

1) It is not clear were amendments to English were done in the paper. In the current form and without extensive English revision, the paper is difficult to read.

3) On lines 216-217 the authors state the ZrO2 particles sizes vary between 20 and 470 nm, as determined by dynamic light scattering. However SEM micrographs show much larger particle aggregates (lines 222-223). The authors should comment on the apparently contradictory results.

Author Response

Revision of Manuscript for Publication

General Remarks

Thank you very much, once again for the opportunity to revise our manuscript. The revised version of the manuscript (Manuscript ID: nanomaterials-676979) titled “Interaction between amorphous zirconia nanoparticles and graphite: Electrochemical applications for gallic acid sensing using carbon paste electrodes in wine” has now been revised and updated. The revised copy has been re-submitted.

We are very appreciative of the thoughtful comments and recommendations. In the manuscript, we have responded to the comments and recommendations appropriately. The revision, changes, and explanations on the manuscript are as follows:-

Response to Reviewer One

General and specific remarks

1) It is not clear were amendments to English were done in the paper. In the current form and without extensive English revision, the paper is difficult to read.

3) On lines 216-217 the authors state the ZrO2 particles sizes vary between 20 and 470 nm, as determined by dynamic light scattering. However, SEM micrographs show much larger particle aggregates (lines 222-223). The authors should comment on the apparently contradictory results.

Response: We are once again very grateful for the patient and meticulous review of the manuscript and he recent feedbacks. Following the valuable advice and comment, the manuscript has been appropriately revised.

Comment 1: 1) It is not clear were amendments to English were done in the paper. In the current form and without extensive English revision, the paper is difficult to read.

Response: We have gone through the manuscript once again and extensively revised the English. All the English revisions have been highlighted in red in the Manuscript.

Comment 2: On lines 216-217 the authors state the ZrO2 particles sizes vary between 20 and 470 nm, as determined by dynamic light scattering. However, SEM micrographs show much larger particle aggregates (lines 222-223). The authors should comment on the apparently contradictory results.

Response: From the results of the Zetasizer particle size analysis the ZrO2 nanoparticles as stated in the manuscript showed sizes that ranged between 20 nm to 470 nm when dispersed in ethylene glycol. Unfortunately, when the powder was analysed using SEM, there seem to have been aggregations of the nanoparticles that made the nanoparticles aggregate on each other and the sizes larger. Hence, we have commented on it in the manuscript in lines 222-226 (highlighted in yellow) thus:- “The increased particle sizes observed in the SEM images could be attributed to any of this aggregation steps (1) adsorption and aggregation of primary building units, small oligomer; (2) formation of thicker oligomers prior to interfacial attachment; (3) lateral growth of nanosheets via addition of building units to edges; and (4) formation of vertical nanosheet stacks accompanied by lateral addition of building units [41]”.